# Virtual Combinatorial Library Screening of Quinadoline B Derivatives against SARS-CoV-2 RNA-Dependent RNA Polymerase

Simone Brogi [1,*], Mark Tristan Quimque [2,3,4], Kin Israel Notarte [5], Jeremiah Gabriel Africa [2], Jenina Beatriz Hernandez [2], Sophia Morgan Tan [6], Vincenzo Calderone [1] and Allan Patrick Macabeo [2,*]

[1] Department of Pharmacy, University of Pisa, Via Bonanno 6, 56126 Pisa, Italy; vincenzo.calderone@unipi.it

[2] Laboratory for Organic Reactivity, Discovery and Synthesis (LORDS), Research Center for the Natural and Applied Sciences, University of Santo Tomas, España Blvd., Manila 1015, Philippines; mjtquimque@gmail.com (M.T.Q.); jeremiahgabriel.africa.sci@ust.edu.ph (J.G.A.); jeninabeatriz.hernandez.sci@ust.edu.ph (J.B.H.)

[3] The Graduate School, University of Santo Tomas, España Blvd., Manila 1015, Philippines

[4] Chemistry Department, College of Science and Mathematics, Mindanao State University—Iligan Institute of Technology, Tibanga, Iligan City 9200, Philippines

[5] Faculty of Medicine and Surgery, University of Santo Tomas, Espana Blvd., Manila 1015, Philippines; kinotarte@gmail.com

[6] Department of Biological Sciences, College of Science, University of Santo Tomas, España Blvd., Manila 1015, Philippines; sophiamorgan.tan.sci@ust.edu.ph

* Correspondence: simone.brogi@unipi.it (S.B.); agmacabeo@ust.edu.ph (A.P.M.)

**Abstract:** The unprecedented global health threat of SARS-CoV-2 has sparked a continued interest in discovering novel anti-COVID-19 agents. To this end, we present here a computer-based protocol for identifying potential compounds targeting RNA-dependent RNA polymerase (RdRp). Starting from our previous study wherein, using a virtual screening campaign, we identified a fumiquinazolinone alkaloid quinadoline B (Q3), an antiviral fungal metabolite with significant activity against SARS-CoV-2 RdRp, we applied in silico combinatorial methodologies for generating and screening a library of anti-SARS-CoV-2 candidates with strong in silico affinity for RdRp. For this study, the quinadoline pharmacophore was subjected to structural iteration, obtaining a Q3-focused library of over 900,000 unique structures. This chemical library was explored to identify binders of RdRp with greater affinity with respect to the starting compound Q3. Coupling this approach with the evaluation of physchem profile, we found 26 compounds with significant affinities for the RdRp binding site. Moreover, top-ranked compounds were submitted to molecular dynamics to evaluate the stability of the systems during a selected time, and to deeply investigate the binding mode of the most promising derivatives. Among the generated structures, five compounds, obtained by inserting nucleotide-like scaffolds (**1**, **2**, and **5**), heterocyclic thiazolyl benzamide moiety (compound **3**), and a peptide residue (compound **4**), exhibited enhanced binding affinity for SARS-CoV-2 RdRp, deserving further investigation as possible antiviral agents. Remarkably, the presented in silico procedure provides a useful computational procedure for hit-to-lead optimization, having implications in anti-SARS-CoV-2 drug discovery and in general in the drug optimization process.

**Keywords:** quinadoline B; SARS-CoV-2; RNA-dependent RNA polymerase inhibitors; virtual screening; combinatorial screening; molecular dynamics

## 1. Introduction

The continued rise in COVID-19 cases worldwide despite the availability of vaccines sustains the demand to discover treatment and prophylactic regimens, particularly through natural products' repurposing and design [1–3]. Computational strategies play a crucial role in accelerating the discovery of effective anti-SARS-CoV-2 agents [4–8], as in silico

experiments are vital in the screening of biologically active compounds, offering a rapid, low-cost, and effective adjunct to in vitro and in vivo experiments. Such methods can facilitate the iteration of known potential compounds to further enhance their biological and pharmacokinetic activities, capable of constructing virtually all possible permutational derivatives from a single parent compound [9].

In COVID-19 drug discovery, several possible drug targets, comprising structural and non-structural proteins, have been exploited in searching novel chemical entities as anti-SARS-CoV-2 agents [10–13]. Among these targets is the RNA-dependent RNA polymerase (RdRp), which is a multi-domain SARS-CoV-2 protein playing a crucial role in the viral life cycle. In particular, RdRp is involved in the replication and transcription of the viral genome [14,15]. Structurally, RdRp is deemed a conserved protein within coronaviruses and carries an accessible region as its active site. Thus, RdRp represents an attractive drug target to inhibit viral replication [14,16]. In our framework, we combined several computational approaches for optimizing a previously described compound targeting SARS-CoV-2 RdRp.

In our recent work, we performed a series of computer-based approaches, employing RdRp as one of the target proteins against fungal secondary metabolites with profound antiviral activity against various known pathogenic viruses. Our work allowed the identification of quinadoline B (Q3, Figure 1), an anti-influenza (H1N1) metabolite isolated from the mangrove-derived fungus *Cladosporium* sp. The fumiquinazoline alkaloid was shown to exhibit a high binding affinity to RdRp, with dynamic stability and favorable pharmacokinetic properties [17]. These results inspired us to further investigate the identified scaffold employing computational drug design methodologies, including structure-based methods such as molecular docking and molecular dynamics, in order to enhance the activity of quinadoline B against SARS-CoV-2 RdRp. Thus, in this study, we structurally redesigned quinadoline B to generate a focused library of derivatives with potentially enhanced antagonism to RdRp through combinatorial in silico techniques.

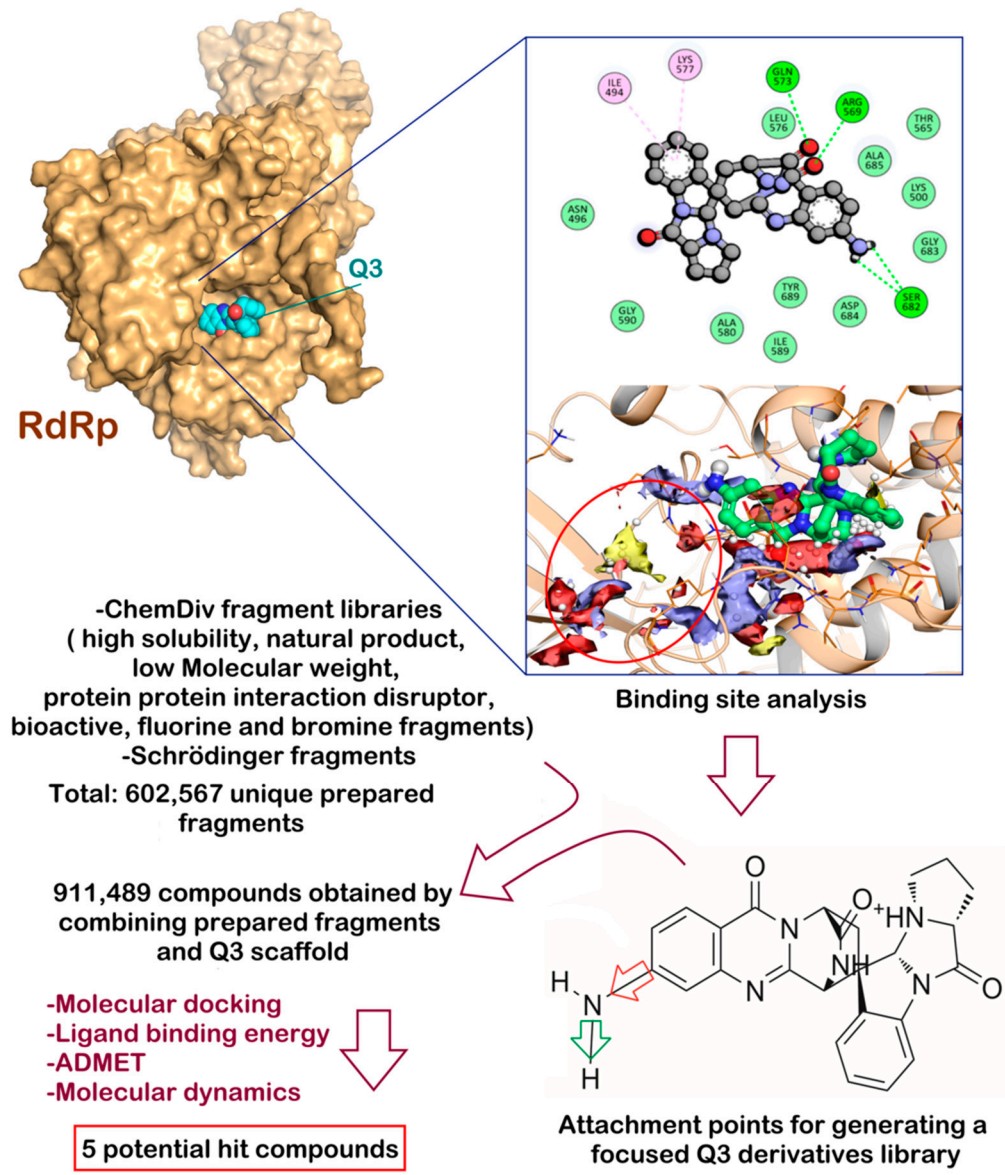

**Figure 1.** Schematic representation of the computational protocol adopted in this study for finding Q3 derivatives with improved in silico affinity for SARS-CoV-2 RdRp.

## 2. Materials and Methods

### 2.1. Computational Details

2.1.1. Ligand and Protein Preparation

Quinadoline B (Q3) was treated by LigPrep (LigPrep release 2018, Schrödinger, LLC, New York, NY, USA, 2018) for identifying the most probable ionization state at cellular pH value (7.4 ± 0.5), and minimized using MacroModel (MacroModel release 2018, Schrödinger, LLC, New York, NY, USA, 2018) implemented in Maestro software (Maestro release 2018, Schrödinger, LLC, New York, NY, USA, 2018), employing OPLS3 as a force field [18]. To simulate the solvent effects, the GB/SA model was employed, selecting "no cutoff" for non-bonded interactions. The PRCG technique (5000 maximum iterations and threshold for gradient convergence = 0.001) was employed to minimize the potential energy.

The structure of the RdRp enzyme of SARS-CoV-2 enzyme was downloaded from the Protein Data Bank (PDB ID 6M71 [19]; crystal structure of RdRp in complex with cofactors) and imported into Maestro suite 2018 and prepared using the protein preparation wizard protocol to acquire an appropriate starting structure for further in silico studies [20,21]. Using this protocol, we performed different computational steps to (1) add hydrogens;

(2) optimize the orientation of hydroxyl groups, Asn, and Gln, as well as the protonation state of His; and (3) perform a constrained minimization refinement using the *impref* utility. At first, the protein was pre-processed by adding all hydrogen atoms to the structure, assigning bond orders, creating disulfide bonds, and filling missing side chains and loops. To optimize the hydrogen bond network, His tautomers and ionization states were predicted; 180° rotations of the terminal angle of Asn, Gln, and His residues were assigned; and hydrogen atoms of the hydroxyl and thiol groups were sampled. Finally, a restrained minimization was performed using the Impact Refinement (*impref*) module, employing an OPLS3 force field to optimize the geometry and minimize the energy of the protein. The minimization was terminated when the energy converged, or the RMSD reached a maximum cutoff of 0.30 Å.

### 2.1.2. Binding Site Analysis

A comprehensive analysis of the binding site of SARS-CoV-2 RdRp was performed using the protein prepared as reported in Section 2.1.1 and the software SiteMap (SiteMap, release 2018, Schrödinger, LLC, New York, NY, USA, 2018).

### 2.1.3. Molecular Docking and Ligand-Energy Evaluation

Glide software (Glide release 2018, Schrödinger, LLC, New York, NY, USA, 2018) employing the XP-scoring function was used to perform all docking studies conducted in this work [22]. The energy grid for docking was prepared using the default value of the protein atom-scaling factor (1.0 Å), with a cubic box centered on the previously identified binding site. The docked poses considered for the post-docking minimization step were 1000, evaluating the Glide XP docking score.

To improve the quality of the screening, we also evaluated the ligand binding energies from the complexes derived by the docking calculation. For this purpose, Prime/MM-GBSA method available in Prime software (Prime release 2018, Schrödinger, LLC, New York, NY, USA, 2018) was used. This technique computes the variation between the free and complex state of both the ligand and enzyme after energy minimization [23,24].

### 2.1.4. Q3-Focused Library Generation

The library was generated as previously reported [25], using several series of fragments obtained from ChemDiv (https://store.chemdiv.com/ accessed on 20 March 2021) in SDF file format. These fragments were treated by LigPrep, in order to convert the 2D structure into the 3D one, and added to Q3 in a side chain hopping approach, considering the selected attachment points that comprise bonds, belonging to the Q3 core structure, replaced in the build process. This strategy allowed to obtain a Q3-focused library that consists of 991,489 compounds. This resulting library was employed in further computational experiments.

### 2.1.5. Evaluation of Drug-like Profile

The drug-like profile was evaluated using SwissADME [26], OSIRIS property explorer, and our in-house cardiotoxicity tool (3D-chERGi) [27]. PAINS assessment was executed employing SwissADME web-server [26], as previously reported [17,28].

### 2.1.6. Molecular Dynamics Simulation Details

Desmond 5.6 academic version, provided by D. E. Shaw Research ("DESRES"), was used to perform MD simulation experiments via Maestro graphical interface (Desmond Molecular Dynamics System, version 5.6, D. E. Shaw Research, New York, NY, USA, 2018. Maestro-Desmond Interoperability Tools, Schrödinger, New York, NY, USA, 2018). MD was performed using the compute unified device architecture (CUDA) API [29] on two NVIDIA GPUs. The complexes derived from docking studies (Figure 2) were imported in Maestro and, using the Desmond system builder, were solvated into an orthorhombic box filled with water, simulated by the TIP3P model [25,30]. An OPLS force field [18] was used for MD calculations. OPLS-aa (all atom) includes every atom explicitly with specific functional groups

and types of molecules, including several biomacromolecules. A distinctive feature of the OPLS parameters is that they were optimized to fit the experimental properties of liquids, such as density and heat of vaporization, in addition to fitting gas-phase torsional profiles. Moreover, it is also largely used by us for performing MD simulations of protein/ligand complexes [25,31,32]. Na$^+$ and Cl$^-$ ions were added to provide a final salt concentration of 0.15 M to simulate the physiological concentration of monovalent ions. Constant temperature (300 K) and pressure (1.01325 bar) were employed with the NPT (constant number of particles, pressure, and temperature) as an ensemble class. RESPA integrator [33] was used to integrate the equations of motion, with an inner time step of 2.0 fs for bonded and non-bonded interactions within the short-range cutoff. Nose–Hoover thermostats [34] were used to maintain the constant simulation temperature, and the Martyna–Tobias–Klein method [35] was applied to control the pressure. Long-range electrostatic interactions were calculated by particle-mesh Ewald method (PME) [36]. The cutoff for van der Waals and short-range electrostatic interactions was set at 9.0 Å. The equilibration of the system was performed using the default protocol provided in Desmond, which consists of a series of restrained minimization and MD simulations applied to slowly relax the system. Consequently, one individual trajectory for each complex of 100 ns was calculated. The trajectory files were analyzed by MD analysis tools implemented in the software package. The same application was used to generate all plots concerning MD simulation presented in this study. Accordingly, the *RMSD* was calculated using the following equation:

$$RMSD_x = \sqrt{\frac{1}{N} \sum_{i=1}^{N} \left( r'_i(t_x) - r_i\left(t_{ref}\right) \right)^2}$$

where the $RMSD_x$ refers to the calculation for a frame *x*; *N* is the number of atoms in the atom selection; $t_{ref}$ is the reference time (typically, the first frame is used as the reference and it is regarded as time *t* = 0); and *r'* is the position of the selected atoms in frame *x*, after superimposing on the reference frame, where frame *x* is recorded at time $t_x$. The procedure is repeated for every frame in the simulation trajectory. Regarding the *RMSF*, the following equation was used for the calculation:

$$RMSF_i = \sqrt{\frac{1}{T} \sum_{t=1}^{T} < \left( r'_i(t) - r_i\left(t_{ref}\right) \right)^2 >}$$

where $RMSF_i$ refers to a generic residue *i*, *T* is the trajectory time over which the *RMSF* is calculated, $t_{ref}$ is the reference time, $r_i$ is the position of residue *i*, *r'* is the position of atoms in residue *i* after superposition on the reference, and the angle brackets indicate that the average of the square distance is taken over the selection of atoms in the residue.

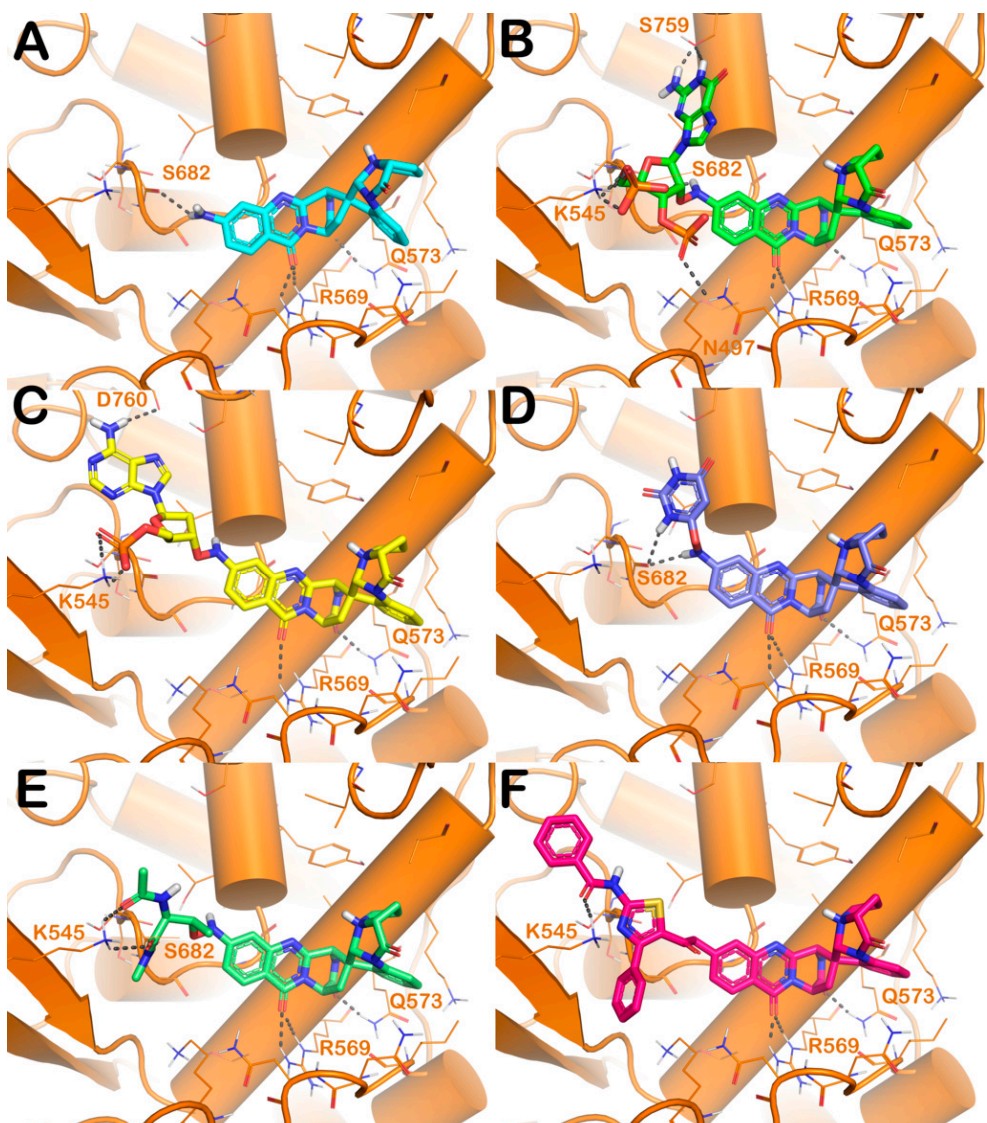

**Figure 2.** Putative binding mode of Q3 (cyan sticks, panel (**A**)) and Q3 derivatives **1–5** (colored sticks, panel (**B**–**F**), respectively) within the SARS-CoV-2 binding site (PDB ID 6M71, orange cartoon). Interacting amino acids are represented by lines, while the H-bonds are indicated by grey-dotted lines. Pictures were generated by PyMOL (The PyMOL Molecular Graphics System, v1.8; Schrödinger, LLC, New York, NY, USA, 2015).

## 3. Results and Discussion

SARS-CoV-2 and its predecessor SARS-CoV have significant similarities in their gene sequence, including the spike (S) glycoprotein, RdRp, and the two cysteine proteases: PL[pro] and 3CL[pro] [37]. Among these viral target proteins, RdRp plays a crucial role in viral replication, and is thus considered an exceptional molecular target for developing anti-SARS-CoV-2 drugs. Accordingly, different fungal derivatives, in particular quinaxoline alkaloids identified from the mangrove-derived fungus *Cladosporium* sp., were identified as possible SARS-CoV-2 RdRp inhibitors [17]. Among them, the ligand quinadoline B (Q3) showed the most interesting inhibitory profile in silico against RdRp. Q3 was found to tightly bind to the active site of RdRp by a series of polar and non-polar interactions. Three H-bonds were observed between the following: (a) the amino group and S682 and (b) carbonyl oxygens of the quinazolinone core and Q573 and R569. The indoline moiety was also involved in π-alkyl interactions with I494 and K577. Several van der Waals interactions against N496, G590, A580, I589, Y689, D684, G683, K500, A685, T565,

and L576 were also noted [17]. The identified binding mode accounted for a binding energy of −9.5 kcal/mol, as found by AutoDock software, highlighting Q3 as one of the most promising derivatives of the series (Figure 1). To further explore the potential of quinadoline B as a drug prototype, in silico combinatorial techniques were employed to generate novel derivatives and enhance the previously reported antagonistic potential to RdRp. To this purpose, we used Schrödinger Drug-discovery Suite. As the first step, we retrieved the previously described binding mode of Q3 within the RdRp binding site by using Glide software (Figure S1). After establishing that the docking protocol was able to correctly locate the quinadoline B scaffold, we deeply investigated the RdRp binding site. The SiteMap analysis revealed the existence of a druggable sub-pocket that can be targeted by modifying Q3 derivatives (Figure 1). In particular, examining the orientation of the compound, we hypothesized that, by introducing appropriate moiety to Q3, possibly linked to the $NH_2$, it could be possible to reach the mentioned sub-pocket at the RdRp binding site. To accomplish this task, we used an in silico structure-based combinatorial library design approach, successfully employed by us, to generate focused libraries targeting specific binding site regions [25]. In the first step, we downloaded several sets of chemical fragments from ChemDiv, including high solubility fragments, natural product fragments, low molecular weight fragments, protein–protein interaction disruptor fragments, bioactive fragments, fluorine and bromine fragments, and other synthetic fragments. These fragments were properly prepared (see Section 2) and added to an existing library available from Schrödinger environment, obtaining 602,567 unique fragments to use in the side chain hopping approach. We selected two possible attachment points on the Q3 derivative exploiting $NH_2$ group (Figure 1). By combining the generated fragments and Q3 at the defined attachment points, we generated a focused library containing 991,489 Q3 derivatives.

The Q3-focused chemical library was employed in a virtual screening protocol based on molecular docking experiments and ligand-binding energy evaluation to identify Q3 derivatives that were able to bind RdRp with greater affinity compared with the starting compound Q3. For this purpose, compounds were docked into the binding site of SARS-CoV-2 RdRp [17] using Glide (Glide release 2018, Schrödinger, LLC, New York, NY, USA, 2018), employing XP as the scoring function and Prime software (Prime release 2018, Schrödinger, LLC, New York, NY, USA, 2018). The output of this step is reported in Table 1. Only Q3 derivatives showing a GlideScore value lower than −6.22 kcal/mol were considered. The threshold was chosen based on the value obtained by performing a docking calculation of Q3 into RdRp. The selected chemical entities were further examined by visual inspection to select molecules displaying a proper binding mode. By employing the above-mentioned computational protocol, we obtained 26 compounds showing improved affinities for the RdRp binding site with respect to the starting compound Q3 (structures are reported in Table S1).

**Table 1.** Final hits and their computational parameters derived from in silico studies.

| Cpd | GlideScore (kcal/mol) | $\Delta G_{bind}$ (kcal/mol) | Main Contacts | LogP$_{o/w}$ [a] | Solubility [b] | GI abs. [c] | PAINS [d] | Tumorigenic [e] | pK$i$ $h$ERG [f] |
|---|---|---|---|---|---|---|---|---|---|
| **1** | −8.71 | −51.1 | H−bonds R569, Q573, S682, N497, S759 salt bridges K545 | −3.72 | High | Low | No | No | 5.03 |
| **2** | −8.47 | −52.3 | H−bonds R569, Q573, K545, D760 | −1.82 | High | Low | No | No | 5.24 |
| **3** | −8.12 | −43.9 | H−bonds R569, Q573, S682 | −0.23 | Moderate | Low | No | No | 5.06 |
| **4** | −7.51 | −44.8 | H−bonds R569, Q573, S682, K545 | −0.27 | Moderate | Low | No | No | 5.35 |
| **5** | −7.46 | −46.3 | H−bonds R569, Q573, K545 cation−π K500, R555 | 3.07 | Poor | Low | No | No | 5.11 |

**Table 1.** *Cont.*

| Cpd | GlideScore (kcal/mol) | $\Delta G_{bind}$ (kcal/mol) | Main Contacts | LogP$_{o/w}$ [a] | Solubility [b] | GI abs. [c] | PAINS [d] | Tumorigenic [e] | pK*i* *h*ERG [f] |
|---|---|---|---|---|---|---|---|---|---|
| **6** | −7.42 | −41.5 | H−bonds R569, Q573, S682 double cation−π K500 | 2.75 | Moderate | High | No | No | 5.32 |
| **7** | −7.38 | −40.6 | H−bonds R569, Q573, S682 double cation−π K500 | 1.67 | Poor | Low | No | No | 5.17 |
| **8** | −7.14 | −41.2 | H−bonds R569, Q573, S682, D684 cation−π K500 | 2.45 | Moderate | High | No | No | 5.51 |
| **9** | −7.08 | −39.1 | H−bonds R569, Q573, S682 | 0.80 | Moderate | Low | No | No | 5.84 |
| **10** | −7.03 | −43.7 | H−bonds R569, Q573, A685, A688 cation−π K545 | 1.32 | Moderate | Low | No | No | 5.63 |
| **11** | −6.97 | −38.8 | H−bonds R569, Q573, S682 cation−π K500 π−π Y689 | 2.50 | Poor | Low | No | No | 4.92 |
| **12** | −6.88 | −40.2 | H−bonds R569, Q573, K545, R555 halogen bonds R624 | 3.02 | Poor | Low | No | No | 5.68 |
| **13** | −6.84 | −42.3 | H−bonds R569, Q573, S682 cation−π K500 | 2.61 | Poor | Low | No | No | 5.26 |
| **14** | −6.81 | −39.4 | H−bonds R569, Q573, S682 cation−π K500 | 1.10 | Moderate | High | No | No | 5.15 |
| **15** | −6.77 | −42.9 | H−bonds R569, Q573, D684 cation−π K545 | 1.05 | Moderate | Low | No | No | 4.93 |
| **16** | −6.71 | −41.0 | H−bonds R569, Q573, S682, K545 | 1.01 | Moderate | High | No | No | 6.21 |
| **17** | −6.59 | −37.1 | H−bonds R569, Q573, S682 π−π Y689 | 2.55 | Poor | Low | No | No | 5.24 |
| **18** | −6.51 | −47.2 | H−bonds R569, Q573, S501 | 1.96 | Poor | Low | No | No | 5.67 |
| **19** | −6.44 | −41.3 | H−bonds R569, Q573, S682 salt bridges D760 | 0.24 | Moderate | High | No | alert: anil_di_alk_A | 5.79 |
| **20** | −6.39 | −33.8 | H−bonds R569, Q573, S682 | 1.84 | Poor | Low | No | No | |
| **21** | −6.37 | −34.9 | H−bonds R569, Q573, S682, K545 | 3.18 | Poor | Low | No | No | 5.47 |
| **22** | −6.36 | −34.3 | H−bonds R569, Q573, S682 halogen bonds K545 | 0.81 | Moderate | Low | No | No | 5.18 |
| **23** | −6.34 | −39.7 | H−bonds R569, Q573 halogen bonds N497 | 1.53 | Poor | Low | No | No | 5.60 |
| **24** | −6.30 | −35.4 | H−bonds R569, Q573, S682 | 0.14 | Moderate | Low | No | No | 5.51 |
| **25** | −6.29 | −40.2 | H−bonds R569, Q573, R553, R555 salt bridges R553, R555 | 1.37 | Moderate | Low | No | No | 4.89 |
| **26** | −6.24 | −39.5 | H−bonds R569, Q573, S682 cation−π K500 | 3.34 | Poor | Low | No | No | 5.54 |
| **Q3** | −6.22 | −32.3 | H−bonds R569, Q573, S682 | −0.12 | Moderate | High | No | No | 5.77 |

[a] Consesus LogP (lipophilicity)—average of five predictions using different algorithms (recommended value < 5); [b] water solubility assessed using three different methods; [c] gastrointestinal (GI) absorption; [d] PAINS (pan-assay interference compounds) predict the possibility of a given compound to behave as PAINS and, consequently, to interfere with biological assay; [e] tumorigenic—the evaluation was performed employing OSIRIS property explorer [38]; [f] predicted activity on seven PLS factors derived from our in-house 3D-QSAR model for predicting *h*ERG K$^+$ channel affinity (3D-chERGi) (pKi (M); pKi > 6, Ki < 1 μM) [27].

The analysis of docking output demonstrated an improvement in the number of contacts (polar and/or hydrophobic contacts) within the selected binding site for all selected compounds along with a greater binding affinity with respect to the starting molecule. The docking results for the five top-ranked compounds are illustrated in Figure 2 in comparison with Q3.

Briefly, starting from compound **1**, obtained by inserting a guanosine-like moiety on Q3 scaffold, we detected the same contacts found for Q3 (H-bonds R569, Q573, and S682) (Figure 2A and Table 1). Additionally, the novel substituent can target the hypothesized region of the RdRp binding site, producing strong interactions with N497, S759, and K545, by polar contacts (Figure 2B and Table 1). This molecular arrangement conferred a strong improvement in binding affinity with respect to the Q3 derivative, showing a GlideScore of −8.71 kcal/mol and a $\Delta G_{bind}$ of −51.1 kcal/mol (Q3, GlideScore −6.22 kcal/mol and $\Delta G_{bind}$ of −32.3 kcal/mol). Interestingly, compound **2** is also modified with a nucleotide moiety. In this case, Q3 was modified by inserting an adenine-like moiety (Figure 2C and Table 1). The docking output revealed that compound **2** similarly interacted within the RdRp binding site compared with compound **1**, except for the lack of H-bonds with N497 and S759 replaced with an H-bond with D760. This strong targeting observation accounted for a significant improvement in the computational score of compound **2** (GlideScore −8.47 kcal/mol and $\Delta G_{bind}$ −52.3 kcal/mol). Compound **3** lacks the previously described contacts, maintaining only the contacts found for Q3 with the addition of an additional H-bond with S682, strongly stabilizing the binding mode (Figure 2D and Table 1), as highlighted by in silico scores (GlideScore −8.12 kcal/mol and $\Delta G_{bind}$ −43.9 kcal/mol) compared with that found for Q3. For compound **4**, the insertion of a peptidic tail allowed to target the residue K545, in addition to the previously described contacts (H-bonds R569, Q573, and S682) (Figure 2E and Table 1). Moreover, in this case, the inserted substituent is well-tolerated by the RdRp binding site, as indicated by the satisfactory computational scores found for compound **4** (GlideScore −7.51 kcal/mol and $\Delta G_{bind}$ −44.8 kcal/mol). Inserting a bulky region with a stronger aromatic nature, as in compound **5**, allowed improvement of hydrophobic contacts within the RdRp binding site. In fact, compound **5** is able to form two cation-π interactions with residues K500 and R555, in addition to the maintained contacts (Figure 2E and Table 1). Compound **5** showed a GlideScore −7.46 kcal/mol and $\Delta G_{bind}$ −46.3 kcal/mol.

To validate the docking output, we conducted MD simulation on the top-five ranked compounds (**1–5**), investigating the evolution of biological systems for 100 ns. In this regard, the resulting trajectories for all complexes were completely examined through different standard simulation parameters including root mean square deviation (RMSD) analysis for all backbone atoms and ligands, and the root mean square fluctuation (RMSF) of individual amino acid residue. The selected complexes showed a general stability from the early stages of the simulation, as indicated by the results found by calculating the RMSD for each complex. In fact, we did not observe any major expansion and/or contraction, after the binding of these compounds during the entire simulation period (Figure 3A–E regarding the simulation of compounds **1–5**, respectively). This stability was also substantiated by observing the RMSF calculated for the selected complexes. RMSF indicates the difference between the atomic Cα coordinates of the protein from its average position during the MD simulation. This calculation is mainly helpful to characterize the flexibility of individual residues in the protein backbone. The considered systems did not show significant fluctuation phenomena, with the exclusion of a restricted number of residues at the N- and C-terminal regions of RdRp (Figure S2). In contrast, the conformational alterations of critical residues in the RdRp binding cleft (lowest RMSF values for all complexes) confirmed the capacity of compounds to form stable interactions within the protein.

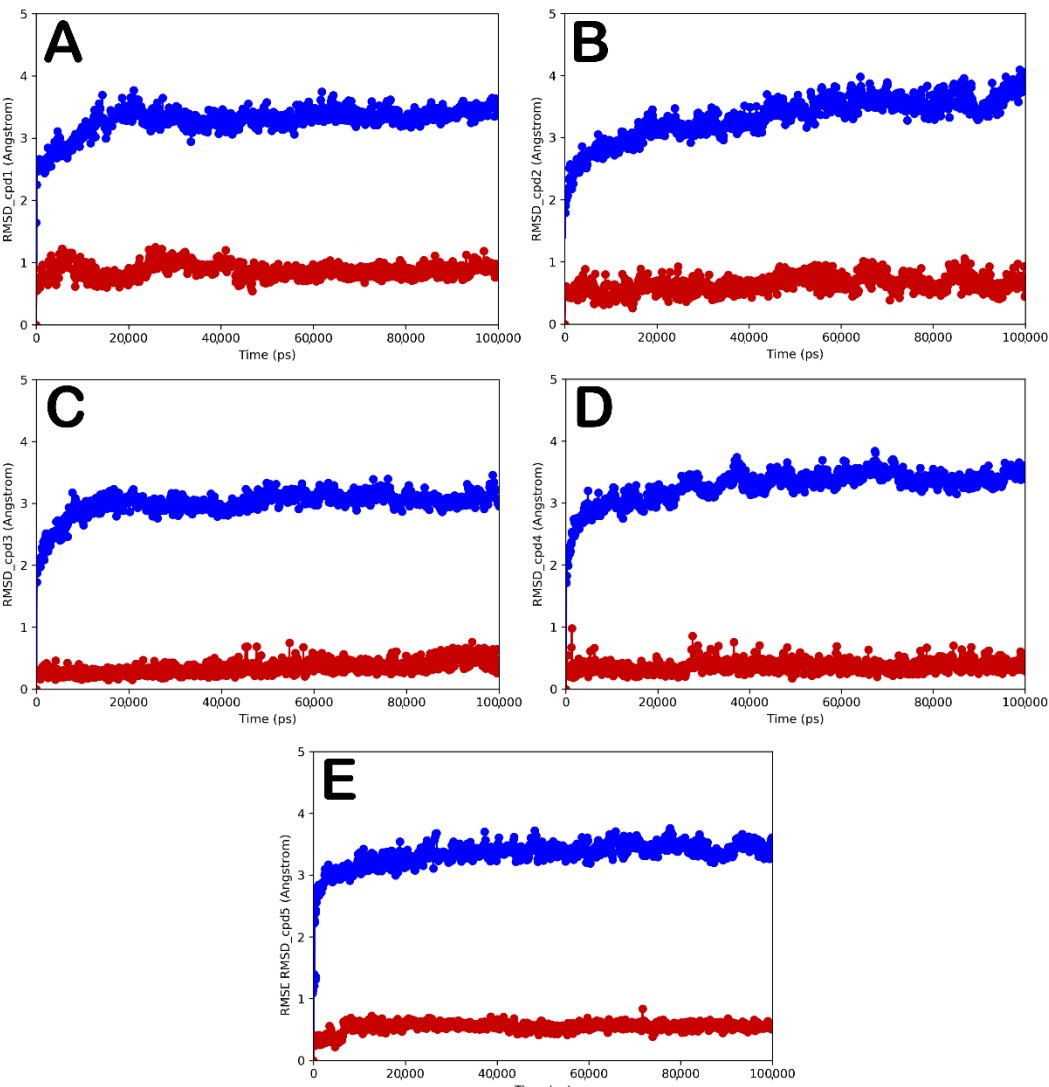

**Figure 3.** RMSD calculation for each complex and for each ligand. RdRp/compounds **1**–**5**, panel (**A**–**E**) (blue line), respectively. Compounds **1**–**5**, panel (**A**–**E**) (red line), respectively.

In order to better understand the behavior of compounds **1**–**5** in the SARS-CoV-2 RdRp binding site, we performed a detailed analysis of the MD simulation investigating the contacts established by compounds in the active site. The output of the analysis performed on the complex RdRp/compound **1** is reported in Figure 4. Compound **1** maintained the contacts found by docking calculation, interacting with R569 and Q573 during the MD simulation, while we observed a decrease in targeting S682. The interactions found by residues N497, S759, and K545 were evident through the time of simulation, as well as the salt bridges. In addition, interactions with A558, T556, R555, and N496 became apparent, while sporadic contacts were observed with residues S681, A685, and D760 considering the 100 ns of the simulation. Analysing the trajectory of compound **2**, we observed that the main contacts established with residues R569, Q573, K545, and D760 were maintained and N496, N497, K500, D623, and S759 were formed, although with no great potency. The output for compound **2** is illustrated in Figure 5. Compound **3** is able to strongly interact with S759 and D760, while less apparent contacts were detected with N496, N497, and D684 in addition to the contacts with the residues R569, Q573, and S682 (Figure 6). The results of this analysis for compounds **4** and **5** are found in the Supplementary Material file (Figures S3 and S4). Compound **4** maintained the contacts through H-bonds with R569, Q573, S682, and K545, while it formed additional contacts with N497, K500, G683, and D684 (Figure S3). Finally, compound **5** was still able to target R569, Q573, K500, and K545, while

the interaction with R555 became sporadic. In contrast, compound **5** strongly targeted N496 and N497 (Figure S3).

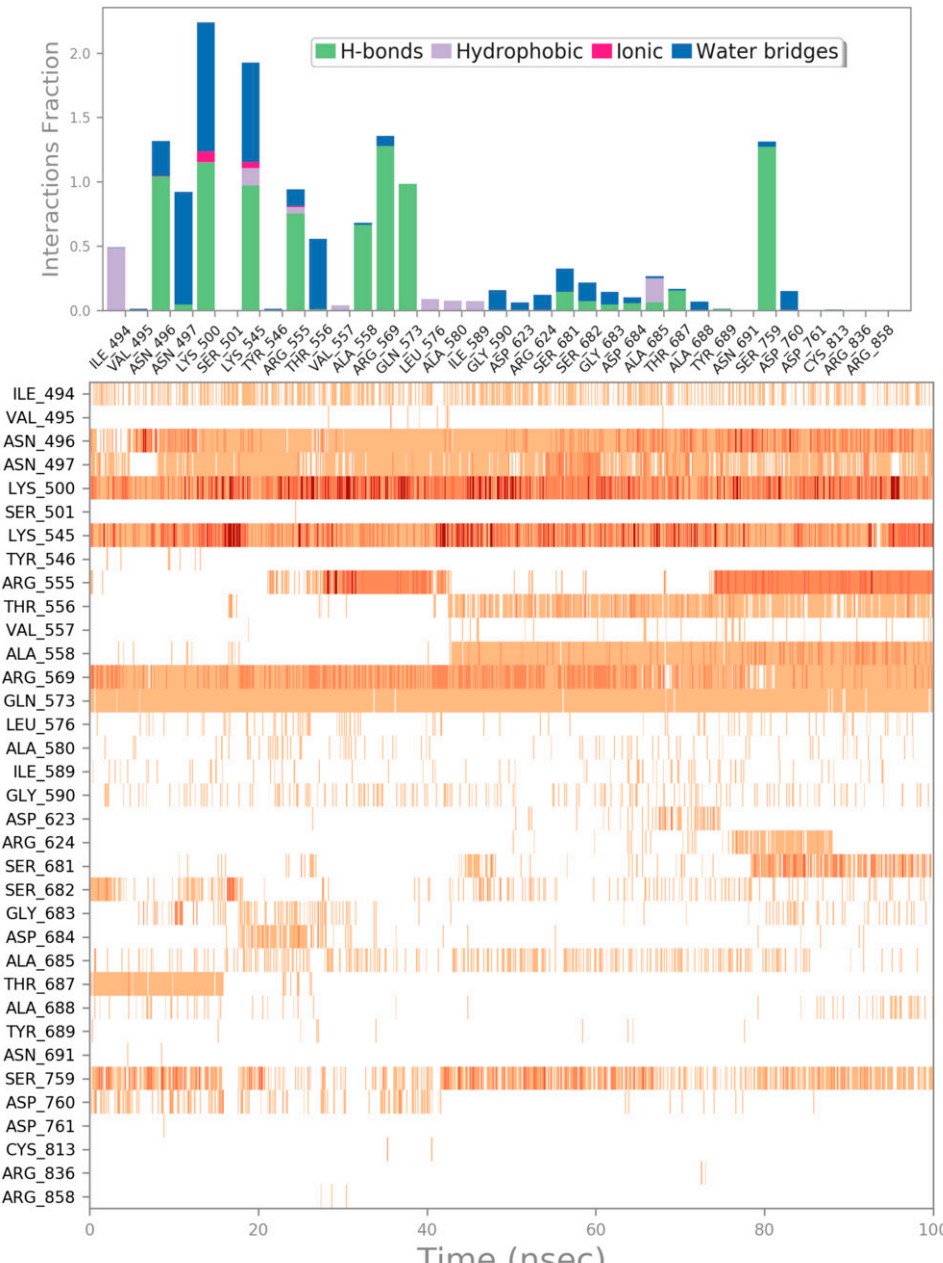

**Figure 4.** Compound **1** monitored during the simulation. The contacts can be grouped by type and summarized, as shown in the plots. Grouping protein–ligand interactions into four types: H-bonds (green), hydrophobic (grey), ionic (magenta), and water bridges (blue). The second graph of the picture displays a timeline representation of the contacts. Some residues make more than one specific contact with the ligand, which is represented by a darker shade of orange.

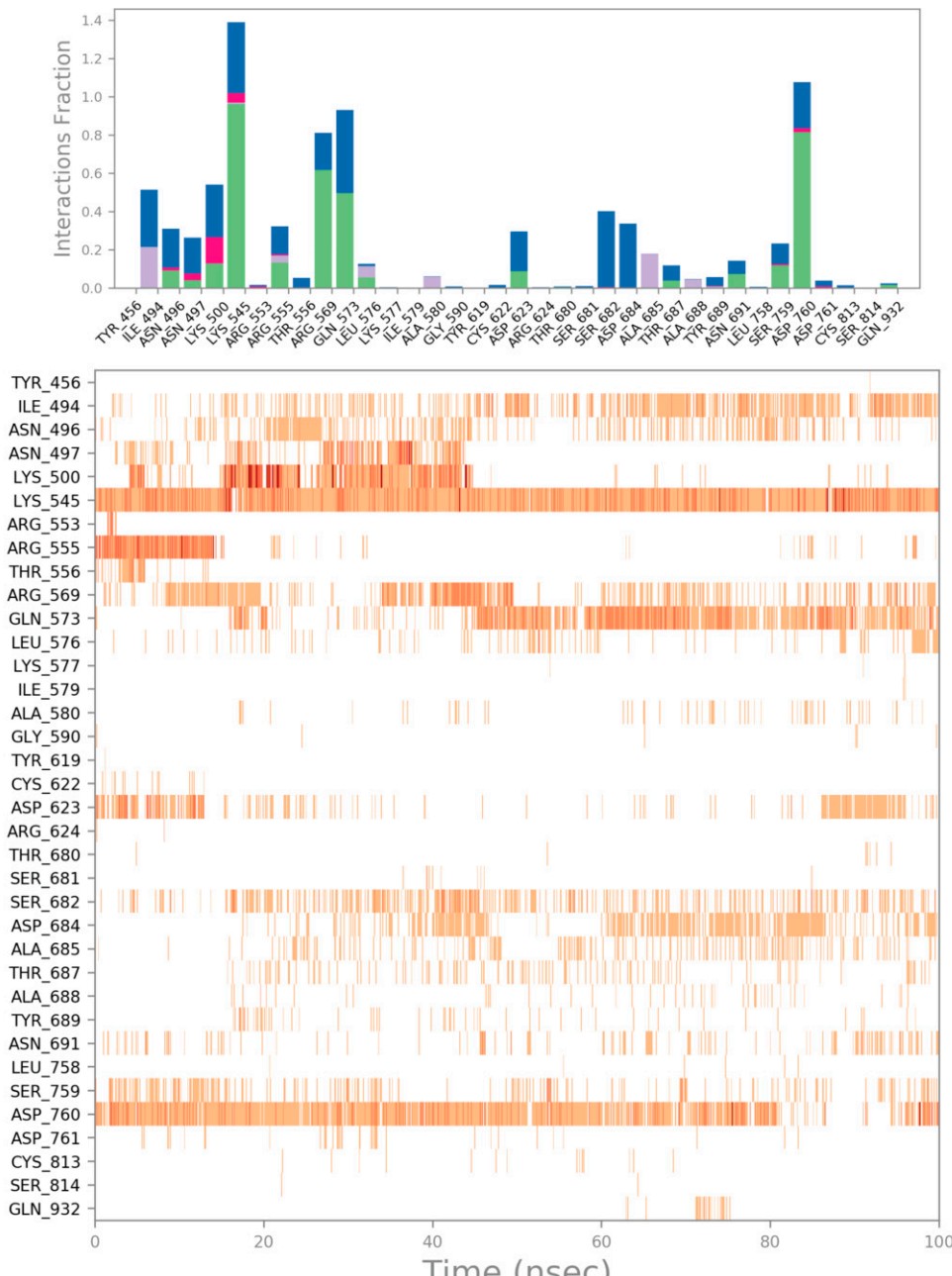

**Figure 5.** Compound **2** monitored during the simulation. The contacts can be grouped by type and summarized, as shown in the plots. Grouping protein–ligand interactions into four types: H-bonds (green), hydrophobic (grey), ionic (magenta), and water bridges (blue). The second graph of the picture displays a timeline representation of the contacts. Some residues make more than one specific contact with the ligand, which is represented by a darker shade of orange.

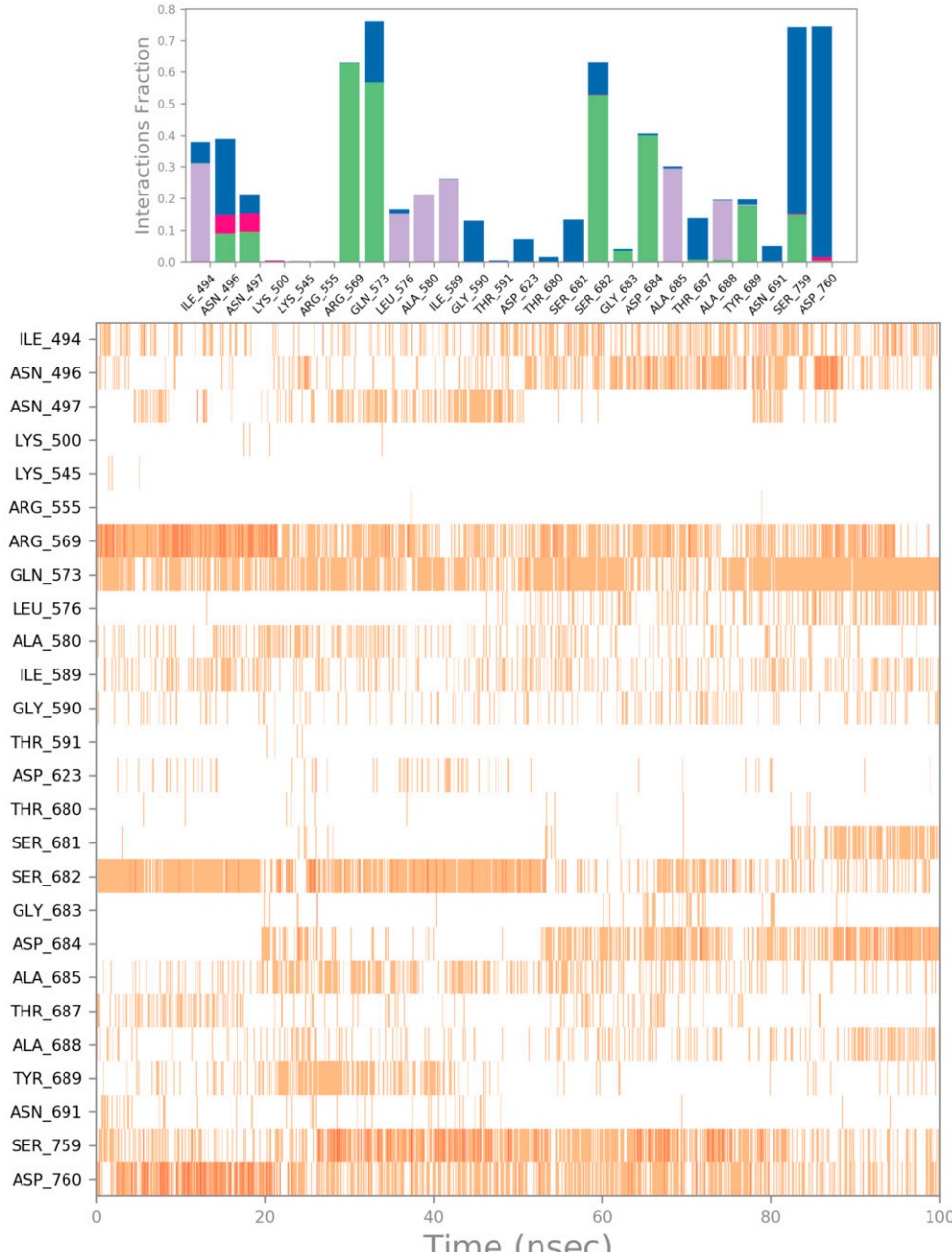

**Figure 6.** Compound **3** monitored during the simulation. The contacts can be grouped by type and summarized, as shown in the plots. Grouping protein–ligand interactions into four types: H-bonds (green), hydrophobic (grey), ionic (magenta), and water bridges (blue). The second graph of the picture displays a timeline representation of the contacts. Some residues make more than one specific contact with the ligand, which is represented by a darker shade of orange.

Overall, the MD simulation outcomes undoubtedly validated the advantageous interactions of five top-ranked compounds screened compounds, showing satisfactory thermodynamic stability in the RdRp binding site, suggesting that they can act as possible SARS-CoV-2 RdRp inhibitors. Furthermore, despite the fact that the addition of bulky moiety results in compounds with a high molecular weight, they showed an acceptable ADMET profile with logP and solubility in acceptable ranges, although the gastrointestinal (GI) absorption was found to be low. They were also found to be non-tumorigenic and devoid of cardiotoxicity, as assessed by our in-house tool, 3D-chERGi [27]; finally, the selected compounds did not have substructural features that allow to behave as pan-assay

interference compounds (PAINS) (Table 1). PAINS compounds are chemical compounds that tend to display activity against numerous targets by nonspecific interactions or by altering the results of the biological tests. Compounds containing such moieties, which are often present in PAINS compounds, could be false positive hits and in general should be removed from the designed series [39]. Accordingly, our computational investigation provided five compounds as potential RdRp inhibitors and, more importantly, suggested guidelines for optimizing compounds considering the binding site of interest, showing improved binding affinity with respect to quinadoline B. In fact, such a structure-based methodology can be easily applied to other ligand–protein complexes for optimizing existing hit compounds.

## 4. Conclusions

In summary, we presented a computer-aided investigation for identifying possible SARS-CoV-2 RdRp inhibitors based on the quinadoline B scaffold, previously identified as possible RdRp ligands [17]. In particular, we used Q3 derivatives to explore the RdRp binding site by inserting several chemical fragments, obtained from the ChemDiv database, obtaining a Q3-focused library of over 900,000 unique structures. This library was used in a virtual screening protocol, employing the crystal structure of SARS-CoV-2 RdRp, for identifying Q3 derivatives with improved binding affinity with respect to quinadoline B. Moreover, the top-ranked compounds were subjected to MD simulations, in order to evaluate the stability of the systems during a selected time, and to deeply investigate the binding mode of the most promising derivatives. Finally, the in silico searching protocol allowed the identification of five compounds with improved affinity for SARS-CoV-2 RdRp, ushering interests for further investigation as possible antiviral agents. Notably, the developed computational protocol has implications in anti-SARS-CoV-2 drug discovery and in general in the drug optimization process, providing a convenient computational procedure for hit-to-lead optimization.

**Supplementary Materials:** The following supporting information can be downloaded at https://www.mdpi.com/article/10.3390/computation10010007/s1, Figure S1: Superposition between the docked pose of Q3 obtained by AutoDock and by Glide into RdRp binding site; Figure S2: RMSF calculation for each complex, selected by docking studies, after 100 ns of MD simulation; Figure S3: Compound **4** monitored during the simulation. The contacts can be grouped by type and summarized, as shown in the plots. Grouping protein–ligand interactions into four types: H-bonds, hydrophobic, ionic, and water bridges; Figure S4: Compound **5** monitored during the simulation. The contacts can be grouped by type and summarized, as shown in the plots. Grouping protein–ligand interactions into four types: H-bonds, hydrophobic, ionic, and water bridges; Table S1: Structure of selected compounds reported as SMILES string.

**Author Contributions:** Conceptualization, S.B., and A.P.M.; methodology, S.B., M.T.Q., K.I.N., V.C., and A.P.M.; software, S.B., M.T.Q., J.G.A., J.B.H., and S.M.T.; validation, S.B., M.T.Q., K.I.N., and A.P.M.; formal analysis, S.B., M.T.Q., K.I.N., V.C., and A.P.M.; investigation, S.B., M.T.Q., K.I.N., J.G.A., J.B.H., S.M.T., V.C., and A.P.M.; writing—original draft preparation, S.B.; writing—review and editing, S.B., M.T.Q., K.I.N., V.C., and A.P.M.; supervision, S.B., and A.P.M. All authors have read and agreed to the published version of the manuscript.

**Funding:** This research received no external funding.

**Institutional Review Board Statement:** Not applicable.

**Informed Consent Statement:** Not applicable.

**Data Availability Statement:** Not applicable.

**Conflicts of Interest:** The authors declare no conflict of interest.

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
