# Peer review of "Virtual Combinatorial Library Screening of Quinadoline B Derivatives against SARS-CoV-2 RNA-Dependent RNA Polymerase"

_computation, doi:10.3390/computation10010007_

Round 1

Reviewer 1 Report

The authors present a computer-based protocol for identifying potential compounds targeting RNA-dependent RNA polymerase (RdRp). They applied an in silico combinatorial methodologies for generating and screening a library of anti-SARS-CoV-2 candidates with strong in silico affinity for RdRp. The authors use the results from a previous study in which a fumiquinazolinone alkaloid quinadoline B (Q3), an antiviral fungal metabolite with significant activity against SARS-CoV-2 RdRp, was identified. The quinadoline pharmacophore was subjected to structural iteration obtaining a Q3-focused library of over 900,000 unique structures. This chemical library was explored to identify binders of RdRp with greater affinity with respect to the starting compound Q3. Five compounds exhibited enhanced binding affinity for SARS-CoV-2 RdRp. It is demonstrated that the presented in silico procedure provides a useful computational procedure for hit-to-lead optimization.

The manuscript is well written. The English is sufficiently good. The manuscript's title, abstract, scheme, tables and figures are adequate to the content.

Author Response

The authors present a computer-based protocol for identifying potential compounds targeting RNA-dependent RNA polymerase (RdRp). They applied an in silico combinatorial methodologies for generating and screening a library of anti-SARS-CoV-2 candidates with strong in silico affinity for RdRp. The authors use the results from a previous study in which a fumiquinazolinone alkaloid quinadoline B (Q3), an antiviral fungal metabolite with significant activity against SARS-CoV-2 RdRp, was identified. The quinadoline pharmacophore was subjected to structural iteration obtaining a Q3-focused library of over 900,000 unique structures. This chemical library was explored to identify binders of RdRp with greater affinity with respect to the starting compound Q3. Five compounds exhibited enhanced binding affinity for SARS-CoV-2 RdRp. It is demonstrated that the presented in silico procedure provides a useful computational procedure for hit-to-lead optimization.

The manuscript is well written. The English is sufficiently good. The manuscript's title, abstract, scheme, tables and figures are adequate to the content.

Authors: we thank the referee for the positive evaluation of the manuscript. A revised version for addressing some minor typos has been submitted.

Reviewer 2 Report

The manuscript entitled "Combinatorial library screening of quinadoline B derivatives against SARS-CoV-2 RNA-dependent RNA polymerase" is an interesting and beneficial theoretical study. Potential compounds targeting RNA-dependent RNA polymerase were assessed in silico.

From the typographic and graphic point of view, the work is processed at a relatively high level. There is a minimum of typos and errors in the work. Please pay attention to the pictures Fig. 4, Fig. 5, Fig. 6 and further Fig. S3 and Fig. S4. The upper part is deformed - the ratio of width and height is not the original one.

The Hyphen symbol (-) is used in the manuscript instead of the correct Minus Sign. This applies to the entire document. Please correct this deficiency.

In Figure 3, the individual points should be presented with smaller circles, due to the fact that the points completely overlap, the informative value of such a representation is relatively low (perhaps the partial transparency of the points would be better, resulting in different shades according to the intensity of the overlap). The reviewer understands that this is probably a common way of presentation, but considers it relatively limited in information.

The question is the similarity index, which is 88% (according to Crossref Similarity Check). This is due to the fact that the work was uploaded by the authors to the ChemRxiv server (https://chemrxiv.org/engage/chemrxiv/article-details/61c1ae9b1e13eb5d2c00a191). How appropriate is this solution in the case of works intended for publication in impact journals is a question. At least two versions are created and some readers then access the article outside the article publisher's website (MDPI), which ultimately damages the publisher. Other matches are minimal, so the work as such is original. Let the editor himself judge this fact and consequences. A document of compliance is attached.

Author Response

The manuscript entitled "Combinatorial library screening of quinadoline B derivatives against SARS-CoV-2 RNA-dependent RNA polymerase" is an interesting and beneficial theoretical study. Potential compounds targeting RNA-dependent RNA polymerase were assessed in silico.

From the typographic and graphic point of view, the work is processed at a relatively high level. There is a minimum of typos and errors in the work. Please pay attention to the pictures Fig. 4, Fig. 5, Fig. 6 and further Fig. S3 and Fig. S4. The upper part is deformed - the ratio of width and height is not the original one.

Authors: we thank the referee for the positive evaluation of the manuscript. A revised version for addressing some minor typos has been submitted. Concerning the figures, we reedited all the pictures to avoid cropping issues.

The Hyphen symbol (-) is used in the manuscript instead of the correct Minus Sign. This applies to the entire document. Please correct this deficiency.

Authors: we replaced Hyphen symbol (-) with minus where necessary through the manuscript.

In Figure 3, the individual points should be presented with smaller circles, due to the fact that the points completely overlap, the informative value of such a representation is relatively low (perhaps the partial transparency of the points would be better, resulting in different shades according to the intensity of the overlap). The reviewer understands that this is probably a common way of presentation, but considers it relatively limited in information.

Authors: we thank the referee for the observation. As argued by the referee, the RMSD graphs in Figure 3 are obtained from the software tools in this way. Unfortunately, it is not possible to change the intensity of the overlap and the transparency of the points, making the picture unmodifiable. Accordingly, we presented the figure as obtained. Moreover, although is present this issue, RMSD graphs are indicative of high stability of the analyzed systems. This method of representation was already used by us in several works (Brogi, S et al 2014 CNS neuroscience & therapeutics 20 (7), 624-632; Sirous, H et al 2019 Computational biology and chemistry 83, 107105; Sirous, H et al Frontiers in chemistry 7, 574; Brogi, S et al 2020 Food & Function 11 (9), 8122-8132; da Silva, ER 2020 Molecules 25 (22), 5271; Sirous, H 2021 Computers in Biology and Medicine 137, 104808; Mondal, S 2021 Computers in Biology and Medicine, 104591 only for citing some examples in which is present the same representation).

The question is the similarity index, which is 88% (according to Crossref Similarity Check). This is due to the fact that the work was uploaded by the authors to the ChemRxiv server (https://chemrxiv.org/engage/chemrxiv/article-details/61c1ae9b1e13eb5d2c00a191). How appropriate is this solution in the case of works intended for publication in impact journals is a question. At least two versions are created and some readers then access the article outside the article publisher's website (MDPI), which ultimately damages the publisher. Other matches are minimal, so the work as such is original. Let the editor himself judge this fact and consequences. A document of compliance is attached.

Authors: we thank the referee for the observation. According to the MDPI policy is possible to upload a preprint version that is not peer-reviewed. In fact, as argued by the referee we submitted the work to ChemRxiv. This allows to obtain an online version of the paper with doi that can be useful for readers and authors, as in this case, since we need to testify our collaboration and the work performed. The upload of the paper to a preprint repository should not be affect the evaluation of the paper.